# Does Perioperative Administration of Rabies Vaccine in Dogs Undergoing Surgical Sterilization Induce an Adequate Antibody Response?

**DOI:** 10.3390/vaccines11091418

**Published:** 2023-08-25

**Authors:** Andrea Peda, Paulina Samaniego, Christy Daugherty, Theresa Wood, Chengming Wang, Darryn Knobel

**Affiliations:** 1Department of Clinical Sciences, Ross University School of Veterinary Medicine, Basseterre P.O. Box 334, Saint Kitts and Nevis; 2Department of Pathobiology, Auburn University, Auburn, AL 36849, USA; woodthe@auburn.edu (T.W.);; 3Department of Biomedical Sciences, Ross University School of Veterinary Medicine, Basseterre P.O. Box 334, Saint Kitts and Nevis

**Keywords:** perioperative vaccination, rabies vaccination, spay/neuter campaigns, sterilization

## Abstract

High-volume spay/neuter events may facilitate access to free-roaming dogs to administer rabies vaccination, but important questions remain regarding the effect of surgery and anesthesia on the immune response to a vaccine administered in the perioperative period. This study evaluated the immunogenicity of primary rabies vaccination in dogs when administered during the immediate perioperative period at the time of surgical sterilization (ovariohysterectomy/orchidectomy). Healthy dogs of both sexes presenting for surgical sterilization who had never been vaccinated against rabies virus were eligible for enrollment in the study. Fifty dogs ranging in age from 5 to 96 months were enrolled and were vaccinated against rabies virus during the recovery period following anesthesia and surgery. Rabies virus neutralizing antibody (RVNA) titers were measured preoperatively and 28 days postoperatively. This cohort was compared to a historical control cohort of 57 dogs who received primary rabies vaccination for travel purposes and had RVNA titers measured at the same laboratory as the study group 28–35 days post-vaccination. After controlling for age and sex, there was no statistically significant difference in immunogenicity of a rabies vaccine administered to dogs during the perioperative period in comparison to dogs that received the rabies vaccine for travel alone in the absence of surgery. Perioperative administration of a rabies vaccine in dogs undergoing surgical sterilization induces an adequate antibody response. We recommend that rabies vaccine be administered perioperatively during spay/neuter campaigns in canine rabies endemic areas if other opportunities to access veterinary care and rabies vaccination are limited.

## 1. Introduction

Rabies caused by infection with *Rabies lyssavirus* transmitted by domestic dogs (*Canis familiaris*) is a neglected but preventable zoonotic disease that predominantly affects populations in resource-limited countries [1]. The World Health Organization (WHO), World Organization for Animal Health (WOAH), Food and Agricultural Organization of the United Nations (FAO), and Global Alliance for Rabies Control (GARC) have endorsed a plan to eliminate human deaths from dog-mediated rabies by 2030 [2]. Elimination of transmission in the reservoir host population through mass vaccination of dogs is an essential component of this plan, and a key short-term outcome is to develop evidence-based tools and strategies for effective dog vaccination [2]. Free-roaming dogs and rabies transmission are integrally linked across many resource-limited countries, and large unmanaged dog populations can be daunting to rabies control program planners [3]. 

Dogs have a global distribution and an estimated total population size of around 700 million, with around 75% considered as free-roaming [4]. Nearly 99% of rabies cases in humans are attributed to dog bites, and free-roaming dogs contribute to maintaining the disease in many countries, particularly in Africa and Asia [2]. The primary focus of rabies control programs in dogs is vaccination; however, rapid population turnover (due to high birth and death rates) of both owned and unowned dogs presents a significant challenge for maintaining high vaccination coverage [5,6]. Although their contribution to rabies control is debatable [3,7,8], mass dog sterilization programs (spay and neuter) supported by governments or non-governmental organizations can clearly reach large numbers of dogs in low-resource areas. By targeting underserved populations for which routine spay-neuter services are unlikely to be available or accessible, these programs provide surgical sterilization to free-roaming dogs or owned dogs that are most at risk of contributing to shelter impoundment and euthanasia [9]. In many cases, dogs in underserved areas would otherwise not receive any routine veterinary care, including vaccinations. Mass dog sterilization programs, therefore, present opportunities to improve rabies vaccination coverage by administering rabies vaccine to dogs around the time of surgical sterilization (perioperatively); however, there is a debate over the immunogenicity of perioperative administration of rabies vaccine.

While the practice of simultaneously vaccinating dogs at the time of surgery or anesthesia is relatively commonplace, it is poorly understood if this practice produces an adequate immune response. Some investigators have reported that surgery and anesthesia can suppress innate and acquired immunity, including impairment of leukocyte trafficking, phagocyte function, mitogenesis, natural killer cell activity, T- and B-cell proliferation, and antibody production [10,11,12,13,14,15]. Other investigators have reported that anesthesia and surgery do not impair responses to vaccination. A perioperative SARS-CoV-2 vaccination guidance document recommended perioperative vaccination against COVID-19 to increase protection from the virus and decrease virus-related mortality in the postoperative period for patients undergoing cardiothoracic surgery. Few studies on the immunogenicity of perioperative rabies vaccination in dogs are available in the literature [16]. One study compared antibody titers following vaccination with an inactivated rabies vaccine and a modified live (ML) canine parvovirus (CPV) vaccine in anesthetized puppies and non-anesthetized controls from the same litters [11]. No surgery was performed in that study. They concluded that the titers of rabies-virus-neutralizing antibodies (RVNAs) were significantly lower in the group of anesthetized animals compared to the control group on the 10th and 20th days post-vaccination, but antibodies to CPV did not differ between the groups [11]. Another study looking at administration of canine distemper virus (CDV) vaccine to mixed-breed female shelter dogs ranging from 4 to 6 months of age during a surgical procedure found that there was no significant adverse effect on antibody response [17]. A study of perioperative vaccination with modified live feline viral rhinotracheitis/calicivirus/panleukopenia (FVRCP) and inactivated rabies virus (RV) vaccines in kittens showed that sterilization at or near the time of first vaccination did not impair antibody responses [10]. Results from another study of perioperative vaccination with modified live FVRCP and inactivated RV in adult cats indicated that vaccination at the time of sterilization appears to induce excellent immune responses, as determined by assessment of serum antiviral antibody titers approximately 10 weeks later [15]. While these studies provide evidence that adult cats and kittens show adequate immunogenicity when vaccinated against rabies with an inactivated monovalent vaccine perioperatively, there is little evidence in the literature to suggest the same in dogs. 

High-volume spay/neuter events may facilitate access to free-roaming dogs to administer rabies vaccination, but important questions remain regarding the effect of surgery and/or anesthesia on the immune response. In this study, we investigated the immunogenicity of perioperative rabies vaccination in previously unvaccinated dogs in the Federation of Saint Kitts and Nevis. We compared the immune response in dogs vaccinated on the same day as elective surgical sterilization (orchidectomy/neuter in males and ovariohysterectomy/spay in females) to that of a historical control group of dogs that did not undergo a surgical or anesthetic procedure around the time of vaccination. Our hypothesis was that the proportion of study subjects adequately responding to vaccination and the mean levels of rabies virus neutralizing antibodies would not differ between the two groups, after adjusting for measured potential confounding variables. 

## 2. Materials and Methods

### 2.1. Study Population

The study population comprised owned, healthy dogs native to the islands of St. Kitts or Nevis in the Caribbean, which had not received a rabies vaccine previously. Health status was determined by a licensed veterinarian through assessment of patient history, physical examination and basic preoperative blood work. Rabies is not considered a core vaccine in the Federation of St. Kitts and Nevis due to the country’s rabies-free status. Dogs undergoing surgical sterilization as part of the Ross University School of Veterinary Medicine (RUSVM) Volunteers for Intercultural and Definitive Adventure (VIDA) club spay and neuter program were eligible for study recruitment. This program conducts spay/neuter days 9 times per calendar year. These events take place at Basseterre Animal Rescue Center (BARC) in St. Kitts and at Nevis Animal Speak (NAS) in Nevis. Owners of dogs presenting for the program between October 2021 and April 2022 were approached to participate in the study and to provide informed consent. One month of flea, tick and heartworm preventive treatment is routinely provided to owners participating in the VIDA spay and neuter program. To compensate participating owners, we offered an additional month of flea, tick and heartworm preventive treatment at the time of the post-vaccination blood collection. Dogs scheduled for a spay or neuter through the VIDA club program have preoperative blood samples routinely drawn to measure packed cell volume (PCV) and test for *Dirofilaria immitis* antigen and antibodies to *Anaplasma phagocytophilum*, *A. platys*, *Borrelia burgdorferi*, *Ehrlichia canis* and *E. ewingii* using the IDEXX SNAP 4Dx test. To be eligible for surgery and enrollment in this study, dogs were required to be negative on the SNAP 4Dx test, have a PCV within an acceptable reference range (37–55%) [18], and be deemed healthy based on history and physical examination. As part of the study, at the time of performing the preoperative blood draw, an additional blood sample was taken from enrolled dogs to measure pre-vaccination rabies-virus-neutralizing antibody (RVNA) levels to confirm unvaccinated status.

### 2.2. Intervention in Treatment Group

Once the preoperative blood draw was completed, patients underwent premedication for anesthesia, anesthetic induction and maintenance, and surgical sterilization. Premedication drugs included a combination of 8 mcg/kg dexmedetomidine and 0.1 mg/kg hydromorphone administered via intramuscular injection. Anesthesia induction was by a combination of 4 mg/kg ketamine and 0.2 mg/kg midazolam administered intravenously to effect. Anesthesia was maintained by administration of 100% oxygen and isoflurane at a level between 1% and 2% by gas anesthesia machine. A ventral midline approach was used for ovariohysterectomy and a prescrotal approach for orchidectomy. Surgeons were licensed veterinarians on the RUSVM faculty. Students in their 7th semester at RUSVM who are members of the VIDA club were able to participate as assistant surgeons and anesthetists as part of their service-learning extracurricular opportunity. The average duration of ovariohysterectomy surgery was 35 min and the average for orchidectomy surgery was 15 min. Local pain management was achieved in males by performing an intratesticular block during patient preparation using 1mg/kg lidocaine and in females by performing an incisional line block at closure using 1 mg/kg lidocaine infused intradermally along the incision line. All dogs received 4.4 mg/kg carprofen administered subcutaneously at completion of the surgery, prior to transfer to the recovery area. In some cases, a repeat dose of hydromorphone at 0.05 mg/kg given subcutaneously was provided 2–4 h after the initial premedication dose for animals experiencing excess pain after initial recovery. Once the patient was stable in the recovery area, a dose of Zoetis Nobivac 3 rabies vaccine was administered subcutaneously over the right hip by a licensed veterinarian. All dogs were monitored closely for 30 min post-vaccination for any signs of hypersensitivity reaction. Patients were sent home with oral carprofen at 4.4 mg/kg to be administered per os once daily for three days starting 24 h after discharge.

### 2.3. Measurement of RVNA Levels

For the treatment group, a pre-vaccination blood sample was drawn on the day of the surgical procedure at the time of the preoperative physical examination. Post-vaccination blood collection was performed 28 days after the procedure. A volume of 5 mL of whole blood was collected and placed in 9 mL red top serum collection tubes. Tubes were placed upright in a rack for a minimum of 30 min and a maximum of 1 h to allow adequate clot formation. Samples were then centrifuged at 3400 rpm for 10 min for serum separation. Serum was removed via pipette and transferred to cryogenic tubes for longer-term freezer storage. Cryotubes were labelled, placed in refrigerator storage at 5 °C for 24 h, and then transferred to freezer storage at −20 °C until completion of follow-up and sample submission to the Auburn University Pathobiology Diagnostic Services (AUPDS) laboratory for RVNA measurement using the rapid fluorescent focus inhibition test (RFFIT) [19]. 

### 2.4. Historical Control Group

The historical control group was confirmed unvaccinated for rabies through medical record assessment. The initial vaccination that was provided was Zoetis Nobivac 3 if given in 2021 and 2022 or Merial Imrab3 if given from 2018 through 2021. These dogs were tested for adequacy of post-vaccination titers 28–35 days after vaccination, for the purposes of temporary travel to the USA and re-importation into St. Kitts. Vaccination and blood collection were performed by veterinarians at the Ross University Veterinary Clinic (RUVC). Measurement of post-vaccination RVNA levels was conducted at the AUPDS laboratory using the RFFIT, as for the treatment group. Data on historical controls were extracted from the RUVC Avimark patient record database by research personnel. Retrospective data were extracted from 2018 to 2022. Only healthy dogs that received a blood draw for titer within 28–35 days after the vaccine were included in the control group. Data extracted included post-vaccination RVNA levels, interval between the vaccine and post-vaccination titer blood draw, age, sex, and breed.

### 2.5. Statistical Analysis

Based on historical data, we estimated that 97% of dogs in the control group would respond adequately to the vaccine. Sample size calculation was based on a test of equivalence at a 5% significance level, 80% power and a prespecified equivalence margin of 10%. Using the SampleSize4ClinicalTrials package in R [20], we obtained a required sample size of 100 (50 in each group).

We evaluated the effect of perioperative vaccination on (i) the proportion of study subjects adequately responding to vaccination and (ii) the RVNA titers obtained, compared to the historical control group. For (i), we used a threshold of 0.5 IU/mL to categorize vaccine response as adequate (post-vaccination titers ≥0.5 IU/mL) or inadequate (<0.5 IU/mL) [21]. The effect of group (perioperative vaccination vs. control) on the risk of inadequate response to vaccination was estimated as the odds ratio from a logistic regression model, controlling for age and sex as potential confounders. We conducted a sensitivity analysis for unmeasured confounders by calculating an E-value, the minimum strength of association that an unmeasured confounder would need to have with both the treatment and the outcome to fully explain away a specific treatment–outcome association, conditional on the measured covariates [22,23]. The effect of group on RVNA titers was estimated from a linear regression model, again controlling for age and sex as potential confounders. For this analysis, RVNA titers were log-transformed to normalize the error distribution. Serum samples that were still not completely neutralizing at the endpoint dilution for the RFFIT were assigned the value in IU/mL at that dilution. In both models, age (in months) was included as a quadratic term. Sex was included as a binary variable (female/male). Statistical analysis was carried out using R statistical software [24].

## 3. Results

In the perioperative vaccine group, 50 dogs were enrolled at 5 VIDA spay/neuter days on St. Kitts and Nevis between October 2021 and April 2022. All 50 dogs had blood collected at enrollment (pre-vaccination) and at 28 days post-vaccination. All subjects had pre-vaccination RVNA levels < 0.1 IU/mL. For the historical control group, complete data from 57 dogs were extracted from the RUVC data management system. All 57 dogs had blood collected 28–35 days post-vaccination and analyzed for RVNA titers at the same laboratory as the perioperative group. Characteristics of the dogs in the two groups and immunogenicity outcomes are shown in Table 1. 

In univariate analyses, perioperative vaccination was associated with an increased risk of failure to adequately respond to vaccination (OR = 5.2, *p* = 0.04) and with decreased mean log-transformed RVNA titers (β = −0.63, *p* = 0.007), compared to the control group. However, after including sex and age in the models to adjust for potential confounding, we found no evidence of an effect of perioperative vaccination on risk of failure (OR = 0.2, *p* = 0.32) or RVNA titers (β = −0.07, *p* = 0.83). Age had a significant positive association with response to vaccine and with RVNA titers. In the sensitivity analysis for unmeasured confounders, we calculated an E-value for the minimum amount of unmeasured confounding needed to move the estimate to a specified odds ratio of 1.1 (that is, a 10% increase in the risk of vaccine failure due to perioperative administration) of 10.5. This means that an unmeasured confounder would have to be associated with a 10-fold increase in the risk of vaccine failure and be 10 times more prevalent in the perioperative group than the historical control group, to explain the observed odds ratio (if the specified odds ratio were true).

## 4. Discussion

We did not find evidence of an effect of perioperative administration on the immunogenicity of rabies vaccine in previously unvaccinated dogs. The proportion of dogs responding adequately to vaccination and RVNA levels obtained 4–5 weeks after vaccination were not statistically significantly different between a cohort of dogs vaccinated during recovery from elective surgical sterilization and a cohort vaccinated for travel purposes, after adjusting for measured confounders (age and sex). All dogs vaccinated perioperatively remained healthy during the 28-day follow-up period.

Other factors may affect the immunogenicity of rabies vaccine. Age was a confounding variable in our study: the median age of dogs in the treatment group was 11 months, whereas in the control group it was 4 years. Age was also significantly associated with immunogenicity, after controlling for group and sex. In a study of dogs vaccinated against rabies for travel to the UK, dogs less than a year old had an increased risk of failure to respond adequately to vaccination [25]. In a study of dogs under one year of age, dogs receiving primary vaccination at ≤16 weeks of age had lower RVNA titers compared to dogs vaccinated at an older age [26]. Although sex distribution differed between our treatment and control groups, this did not affect our results as sex was not associated with vaccine immunogenicity, after controlling for age and group. Mansfield et al. [25] and Wallace et al. [26] also found that the risk of rabies vaccine failure was not associated with sex in dogs. Other studies have found that measures of immunogenicity are affected by the interval between vaccination and measurement of RVNA titers [25,26] and by health status [27,28]. To control for these effects, we restricted our study to measurement of RVNA titers collected from 4 to 5 weeks after primary vaccination in apparently healthy patients with no known underlying conditions.

The primary limitation of our study was the use of an historical rather than contemporaneous control cohort. An ideal randomized control trial (‘target trial’, [29]) would have assigned eligible dogs to vaccine only (control group) or vaccine plus surgery (perioperative vaccine group) and measured immunogenicity outcomes in both groups simultaneously. We attempted to emulate this target trial by adjusting for measured confounders and by restricting our study subjects, as described above. Furthermore, we restricted our study to outcomes in both groups measured in the same laboratory using the same standard diagnostic test (RFFIT). Although there is still potential for bias due to unmeasured confounding in our results (e.g., change in vaccine potency or diagnostic method over time), effect estimates are robust to unmeasured confounding and in our opinion are generalizable/transportable to other similar populations of interest (primary vaccination of dogs of either sex undergoing elective surgical sterilization after passing a general health screen).

In conclusion, we found that perioperative administration of rabies vaccine in dogs undergoing surgical sterilization induces an adequate antibody response. We recommend that rabies vaccine be administered perioperatively during spay/neuter campaigns in underserved communities in canine-rabies-endemic areas if other opportunities for access to veterinary care and rabies vaccination are limited. In all situations, it is recommended that dogs receive a booster vaccination for rabies within 12 months from their initial dose to ensure an adequate antibody response.

## Figures and Tables

**Table 1 vaccines-11-01418-t001:** Characteristics of the dogs in the two groups and immunogenicity outcomes.

Characteristic	Perioperative Vaccine	Historical Control
Group size	50	57
Sex		
Female	31 (62%)	30 (53%)
Male	19 (38%)	27 (47%)
Age in months, median (IQR)	11 (7–27)	48 (48–60)
Days sampled post-vaccination, mean (standard deviation)	28 (0)	31.5 (2.15)
RVNA titers in IU/mL, geometric mean (geometric standard deviation)	1.35 (3.01)	2.55 (3.44)
Proportion with post-vaccination RVNA titers ≥ 0.5 IU/mL	84%	96%

## Data Availability

The data presented in this study are available on request from the corresponding author.

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
