# Peer review of "Does Perioperative Administration of Rabies Vaccine in Dogs Undergoing Surgical Sterilization Induce an Adequate Antibody Response?"

_vaccines, 2023, doi:10.3390/vaccines11091418_

Round 1
Reviewer 1 Report
In the manuscript submitted the authors address the question “Does perioperative administration of rabies vaccine in dogs undergoing surgical sterilization induce an adequate antibody response?.” The question has not definitively been answered previously. As stated in the paper, the global goal of elimination of human deaths caused by dog rabies can be advanced by vaccinating dogs upon sterilization procedures already established. The paper is very well-written. Methods are clearly defined; however some clarification is needed as described below. The limitation of use of historical data and the associated downsides with use adjustments applied was adequately described.
Lines 176 to 189: regarding the RFFIT result values used in statistical analysis, are ‘titers’ the endpoint dilution obtained, or the IU/mL values obtained by calculation from the endpoint dilution values? Because the threshold value is given in IU/mL (0.5 IU/mL) and the upper limit given in endpoint dilution (1:625), description of the relationship between the values would be useful. It is unclear whether 1:625 was assigned or the IU/mL associated with 1:625 was assigned and if the IU/mL was a constant or variable value.
Minor comments:
Line 154: Reference for the RFFIT procedure used is needed.
Line 179: A suggested reference for the 0.5 IU/mL is the Rabies Chapter (31.18) in the WOAH Terrestrial Manual of Laboratory https://www.woah.org/fileadmin/Home/eng/Health_standards/tahm/3.01.18_RABIES.pdf
Author Response
In the manuscript submitted the authors address the question “Does perioperative administration of rabies vaccine in dogs undergoing surgical sterilization induce an adequate antibody response?.” The question has not definitively been answered previously. As stated in the paper, the global goal of elimination of human deaths caused by dog rabies can be advanced by vaccinating dogs upon sterilization procedures already established. The paper is very well-written. Methods are clearly defined; however some clarification is needed as described below. The limitation of use of historical data and the associated downsides with use adjustments applied was adequately described.
Lines 176 to 189: regarding the RFFIT result values used in statistical analysis, are ‘titers’ the endpoint dilution obtained, or the IU/mL values obtained by calculation from the endpoint dilution values? Because the threshold value is given in IU/mL (0.5 IU/mL) and the upper limit given in endpoint dilution (1:625), description of the relationship between the values would be useful. It is unclear whether 1:625 was assigned or the IU/mL associated with 1:625 was assigned and if the IU/mL was a constant or variable value.
RESPONSE: This sentence has been rewritten as follows in an effort to improve clarity: “Serum samples that were still not completely neutralizing at the endpoint dilution for the RFFIT were assigned the value in IU/mL at that dilution” (lines 198-199)
Minor comments:
Line 154: Reference for the RFFIT procedure used is needed.
RESPONSE: A reference for the RFFIT procedure has been provided
Line 179: A suggested reference for the 0.5 IU/mL is the Rabies Chapter (31.18) in the WOAH Terrestrial Manual of Laboratory https://www.woah.org/fileadmin/Home/eng/Health_standards/tahm/3.01.18_RABIES.pdf
RESPONSE: The suggested reference is the one used
Reviewer 2 Report
The manuscript “Does perioperative administration of rabies vaccine in dogs undergoing surgical sterilization induce an adequate antibody response?” presents interesting findings on the immunogenicity of perioperative rabies vaccination in dogs undergoing surgical sterilization. However, the study has several limitations. Addressing these limitations would enhance the strength and validity of the study's conclusions.
The introduction briefly mentions that there is a debate over the immunogenicity of perioperative administration of rabies vaccine in dogs, but it does not explicitly state the research gap or the specific question that the study aims to address. The introduction should clearly outline the knowledge gap and why it is essential to investigate the immune response to perioperative rabies vaccination in dogs. While the introduction refers a few studies related to perioperative vaccination in dogs and other animals, it could benefit from a more comprehensive review of the literature. Including more recent and relevant studies in the field would strengthen the background information and support the study's rationale. Furthermore, the introduction mentions the global distribution of dogs and their role in rabies transmission, but it does not provide a clear contextualization of the specific geographical area or population the study aims to investigate. Providing this context would make the study's objectives more relevant and meaningful. Additionally, the introduction lacks a clear and specific research hypothesis that the study aims to test. The authors should state their hypothesis in a concise manner to guide readers and provide a framework for the study.
The methodology section does not mention whether the selection of dogs for the perioperative group and the historical control group was randomized. Randomization is crucial to ensure that the two groups are comparable in terms of relevant variables, reducing potential biases. The study appears to use a historical control group for comparison. Historical controls may not adequately control for confounding factors and biases. A randomized controlled trial design with a contemporaneous control group would strengthen the validity of the study. Additionally, the study accounts for age and sex as potential confounders in the logistic regression and linear regression models. However, other factors like breed, weight, health status, and environmental exposures might also influence the vaccine response. The inclusion of additional relevant covariates and their impact on the results should be addressed. Line No. 191 states “Statistical analysis was done using R statistical software (reference)”. The authors should add the relevant citation. The manuscript does not specify the type of rabies vaccine used in the perioperative group or the historical control group. Different vaccines might have varying efficacy levels, leading to potential discrepancies in immunogenicity outcomes.
Minor editing required
Author Response
The manuscript “Does perioperative administration of rabies vaccine in dogs undergoing surgical sterilization induce an adequate antibody response?” presents interesting findings on the immunogenicity of perioperative rabies vaccination in dogs undergoing surgical sterilization. However, the study has several limitations. Addressing these limitations would enhance the strength and validity of the study's conclusions.
The introduction briefly mentions that there is a debate over the immunogenicity of perioperative administration of rabies vaccine in dogs, but it does not explicitly state the research gap or the specific question that the study aims to address.
RESPONSE: The specific question that the study aims to address is provided in the title of the manuscript, “Does perioperative administration of rabies vaccine in dogs undergoing surgical sterilization induce an adequate antibody response?”
The introduction should clearly outline the knowledge gap and why it is essential to investigate the immune response to perioperative rabies vaccination in dogs.
RESPONSE: The knowledge gap is made explicit in the Introduction in lines 86-89, “While these studies support evidence that adult cats and kittens show adequate immunogenicity when vaccinated against rabies with an inactivated monovalent vaccine perioperatively, there is little evidence in the literature to suggest the same in dogs” and again in lines 91-92, “High volume spay/neuter events may facilitate access to free-roaming dogs for rabies vaccination, but important questions remain regarding the effect of surgery and/or anesthesia on immune response.”
While the introduction refers a few studies related to perioperative vaccination in dogs and other animals, it could benefit from a more comprehensive review of the literature. Including more recent and relevant studies in the field would strengthen the background information and support the study's rationale.
RESPONSE: We wish to bring to the reviewer’s attention that this manuscript is submitted as a “Brief Report”. Within those constraints, we have provided a comprehensive background to frame the research question including a review of the published literature related to perioperative vaccination in dogs and other animals, which is not extensive (lines 59-89). If the reviewer is aware of more recent and relevant studies on immunogenicity of rabies vaccine administered perioperatively in dogs, we would be grateful to receive those references and review them for inclusion.
Furthermore, the introduction mentions the global distribution of dogs and their role in rabies transmission, but it does not provide a clear contextualization of the specific geographical area or population the study aims to investigate. Providing this context would make the study's objectives more relevant and meaningful.
RESPONSE: The specific geographical area of the study is now added in the Introduction (lines 93-94).
Additionally, the introduction lacks a clear and specific research hypothesis that the study aims to test. The authors should state their hypothesis in a concise manner to guide readers and provide a framework for the study.
RESPONSE: The research hypothesis is now stated in the Introduction (lines 97-100).
The methodology section does not mention whether the selection of dogs for the perioperative group and the historical control group was randomized. Randomization is crucial to ensure that the two groups are comparable in terms of relevant variables, reducing potential biases. The study appears to use a historical control group for comparison. Historical controls may not adequately control for confounding factors and biases. A randomized controlled trial design with a contemporaneous control group would strengthen the validity of the study. Additionally, the study accounts for age and sex as potential confounders in the logistic regression and linear regression models. However, other factors like breed, weight, health status, and environmental exposures might also influence the vaccine response. The inclusion of additional relevant covariates and their impact on the results should be addressed.
RESPONSE: Dogs were not randomly assigned to the perioperative group or the historical control group. We agree that a randomized trial with a contemporaneous control group would have been preferable and in fact address that limitation of our study by discussing the concept of ‘target trial emulation’ (lines 250-255) and describing the steps taken (adjustment for measured confounding and restriction for health status) to improve validity of our estimates. We agree that other factors like breed, weight and environmental exposures may be unmeasured confounders; however, our estimate of the E-Value (presented in lines 218-225) suggests that our effect estimates are robust to unmeasured confounding (an unmeasured confounder would have to be associated with a 10-fold increase in the risk of vaccine failure and be 10 times more prevalent in the perioperative group than the historical control group).
Line No. 191 states “Statistical analysis was done using R statistical software (reference)”. The authors should add the relevant citation.
RESPONSE: The relevant citation has been added
The manuscript does not specify the type of rabies vaccine used in the perioperative group or the historical control group. Different vaccines might have varying efficacy levels, leading to potential discrepancies in immunogenicity outcomes.
RESPONSE: The types of rabies vaccines are specified in lines 147 (perioperative group) and lines 167-168 (historical control group).
Reviewer 3 Report
Congratulations to the authors for their well written and illustrative manuscript. Just a few brief observations.
Although the control group was not contemporaneous with the treatment group, the authors correctly discuss and argue the case.
Line 120. Include a valid reference for the range values considered.
Line 140. Briefly describes the post-surgical management, drugs used, doses, etc.
Line 154. Can authors briefly describe the RFFIT methodology?
Line 191 Include a reference at the end of the paragraph
Author Response
Congratulations to the authors for their well written and illustrative manuscript. Just a few brief observations.
Although the control group was not contemporaneous with the treatment group, the authors correctly discuss and argue the case.
Line 120. Include a valid reference for the range values considered.
RESPONSE: A reference for the range values considered is now provided.
Line 140. Briefly describes the post-surgical management, drugs used, doses, etc.
RESPONSE: A brief description of the post-surgical management, including drugs used and doses, is now provided (lines 141-151)
Line 154. Can authors briefly describe the RFFIT methodology?
RESPONSE: A reference is now provided to the RFFIT methodology
Line 191 Include a reference at the end of the paragraph
RESPONSE: A reference is now included at the end of the paragraph.
Round 2
Reviewer 2 Report
I am glad to know that the authors have addressed the suggested changes and have significantly improved the manuscript. It would be my pleasure to recommend article for publication.